# Learning Morphological Feature Perturbations for Calibrated Semi-Supervised Segmentation

**Mou-Cheng Xu**[1,2]                                    MOUCHENG.XU.18@UCL.AC.UK
**Yu-Kun Zhou**[1,2]                                     YUKUN.ZHOU.19@UCL.AC.UK
**Chen Jin**[1,3]                                         JIN.CHEN@UCL.AC.UK
**Stefano B. Blumberg**[1,3]                            STEFANO.BLUMBERG.17@UCL.AC.UK
**Frederick J. Wilson**[4]                               FRED.WILSON@PHYSICS.ORG
**Marius de Groot**[4]                                   MARIUS.X.DE-GROOT@GSK.COM
**Daniel C. Alexander**[1,3]                            D.ALEXANDER@UCL.AC.UK
**Neil P. Oxtoby**[*,1,3]                                N.OXTOBY@UCL.AC.UK
**Joseph Jacob**[*,1,5]                                  J.JACOB@UCL.AC.UK

[1] *Centre For Medical Image Computing, UCL, UK*

[2] *Department of Medical Physics and Biomedical Engineering, UCL, UK*

[3] *Department of Computer Science, UCL, UK*

[4] *GlaxoSmithKline Research & Development, Stevenage, UK*

[5] *UCL Respiratory, University College London Hospital, UCL, UK*

[*] *Joint Senior Authorship*

**Editors:** Under Review for MIDL 2022

## Abstract

We propose MisMatch, a novel consistency-driven semi-supervised segmentation framework which produces predictions that are invariant to learnt feature perturbations. MisMatch consists of an encoder and a two-head decoder. One decoder pays positive attention to the foreground regions of interest (RoI) on unlabelled images thereby learning dilated features. The other decoder pays negative attention to the foreground on the same unlabelled images thereby learning eroded features. We then apply a consistency regularisation on the paired predictions. MisMatch outperforms state-of-the-art semi-supervised methods on a CT-based pulmonary vessel segmentation task and a MRI-based brain tumour segmentation task. In addition, we show that the effectiveness of MisMatch comes from better model calibration than its fully supervised learning counterpart. Code can be found here: https://github.com/moucheng2017/Learning_Morphological_Perturbation_SSL

**Keywords:** Semi-Supervised, Learning Augmentation, Differentiable Morphological Operations, Attention, Calibration, Consistency Regularisation, Segmentation

## 1. Introduction

Medical image segmentation using deep learning requires expertly-labelled big data. Labels are scarce because manual labelling of medical images by experts is prohibitively expensive in both time and money. Semi-supervised learning (SSL) aims to tackle label scarcity by leveraging information in the unlabelled data. To date, SSL has relied on two key assumptions: the cluster assumption and the smoothness assumption. The cluster assumption states that data points belonging to the same cluster are more likely to be in the same class (Cahpelle et al., 2006). The smoothness assumption presumes that data points are

more dense in the centre of a cluster. According to the cluster and smoothness assumptions, the optimal decision boundary should lie in a low-density region between clusters of data points. Consistency regularisation (Tarvainen and Valpola, 2017) with image-level perturbations can locate the decision boundary for image classification tasks, but does not generalise to image segmentation (Fig. 1).

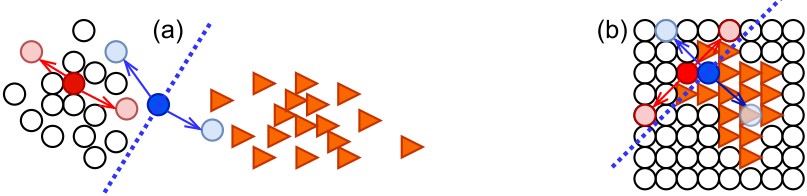

Figure 1: Illustration of consistency regularisation based SSL with data augmentation for: (a) classification; (b) segmentation. In (a), image-level perturbations with consistency learns a decision boundary in a low-density region, which can separate data points (images). This does not generalise to pixel-level data points: the decision boundary in (b) does not lie in a low-density region, and cannot separate data points (pixels).

**Image-level Consistency Regularisation in Classification.** Consistency regularisation has achieved state-of-the-art performance across different SSL image classification tasks, by forcing the model to produce perturbation-invariant predictions (Tarvainen and Valpola, 2017; Sohn et al., 2020; Berthelot et al., 2020; Athiwaratkun et al., 2019). This is illustrated in the cartoon of Fig. 1(a) where each data point is an image. Let's define a perturbation on a data point as randomly changing its position in the space. A consistency loss (e.g. mean-squared error) is defined as the difference between the predictions on two different perturbations of one data point. Fig. 1(a) shows two different perturbations (red arrows) applied on the red data point lying in the high-density region, resulting in zero consistency loss as neither of the red perturbed data points crosses the decision boundary. On the other hand, when two different perturbations (blue arrows) are applied on the blue data point lying in the low-density region, the two corresponding predictions on the two perturbed blue data points will be different, leading to a valid consistency loss value which drives the decision boundary to stay in the low-density region.

**Pixel-level Consistency Regularisation in Segmentation.** Image segmentation operates at the pixel level, where the concept of a low-density region does not exist because pixels are uniformly and densely distributed. Hence, the assumptions behind consistency-based SSL are invalid for segmentation at the pixel level (French et al., 2020). This is illustrated in the cartoon of Fig. 1(b), where the decision boundary does not correspond to the edge of the region of interest, making segmentation difficult/inaccurate. Fortunately, (Ouali et al., 2020) reported that low-density regions can be observed at the feature level and, more importantly, that low-density regions align well with the optimal decision boundaries, i.e., edges of regions of interest.

**Feature-level Consistency Regularisation for Segmentation.** Following the intuition above (see Fig. 1), appropriate feature-level perturbations should cross low-density

regions in feature space, corresponding to the periphery of the foreground regions of interests. Naturally, we are inspired by the classic morphological operations dilation and erosion, which respectively add or remove boundary pixels to a given region of interest while preserving its shape. However, classic morphological operations are not differentiable thereby not suitable for being integrated into deep learning models. In Sec. 3.1, we show a baseline which straightforwardly applies morphological operations on the features are not ideal for consistency regularisation. A previous approach proposed hand-crafted feature perturbations (Ouali et al., 2020), which do not generalise well and not differentiable.

In this paper we introduce MisMatch, a deep, end-to-end framework for semi-supervised segmentation with consistency regularisation. The key novelty of MisMatch is to avoid the non-differentiable hand-crafted perturbations by using attention mechanisms to learn differentiable morphological perturbations at the feature level directly from the data.

## 2. Methods

The overarching concept behind MisMatch is to leverage different attention mechanisms to respectively dilate and erode the foreground features, which are combined in a consistency-driven framework for semi-supervised segmentation. As shown in Fig.2, MisMatch is a framework which can be integrated into any encoder-decoder based segmentation architecture.

**Shape-Constrained Morphological Operations At Feature Level.** Whereas classical morphological operations change the boundary of the foreground at the image level and not differentiable, our network topology is designed to learn to morph the features. We combine two concepts. First, results in (Wei et al., 2018; Chen et al., 2017; Luo et al., 2016; Xu et al., 2020b) showing that Atrous convolution can enlarge foreground features by increasing false positives on the foreground boundary. Second, results in (Luo et al., 2016; Xu et al., 2020b) showing that skip-connections can shrink foreground features. In combination, we can achieve learning-based feature perturbations with both Atrous convolution and skip-connections for consistency-driven semi-supervised segmentation.

**Architecture of MisMatch.** We use U-net (Ronneberger et al., 2015) as backbone due to its popularity in medical imaging. Our MisMatch (**Fig 2**) has two components: an encoder ($f_e$) and a two-head decoder ($f_{d1}$ and $f_{d2}$). The first decoder ($f_{d1}$) comprises a series of *Positive Attention Shifting Blocks*, which dilates the foreground. The second decoder ($f_{d2}$) contains a series of *Negative Attention Shifting Blocks*, which erodes the foreground. The details of the architecture is in Fig.6.

### 2.1. Positive Attention Shifting Block in $f_{d1}$

The Positive Attention Shifting Block (PASB) (Pink block in Fig.6) dilates foreground features with positive attention to the foreground. Each PASB has two parallel branches, namely the main branch and the side branch. The main branch is used for processing visual information and it has the same architecture with a decoder block in a standard U-net, which comprises two consecutive convolutional layers with kernel size 3 followed by ReLU and normalisation layers. The side branch is used to generate a dilating attention mask to guide the main branch to enlarge its foreground features. To do so, the side branch uses two consecutive Atrous convolutional layers with kernel size 3 and dilation rate at 5,

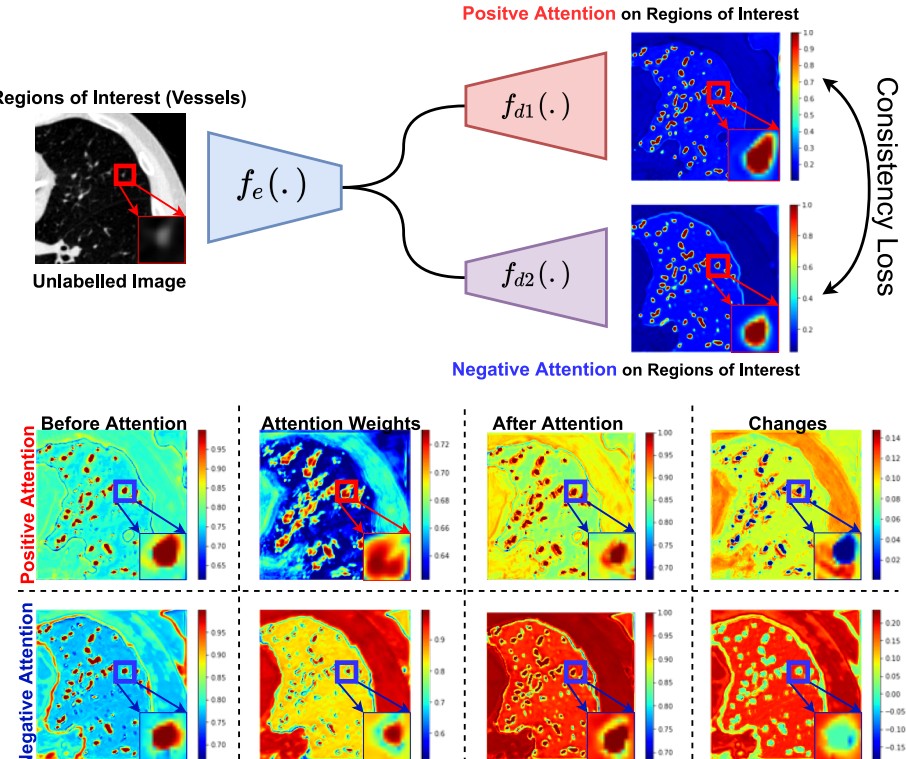

Figure 2: MisMatch learns confidence-invariant predictions on the foreground: decoder $f_{d1}$ dilates foreground features and decoder $f_{d2}$ erodes foreground features. The final prediction is the average between outputs of $f_{d1}$ and $f_{d2}$. Any encoder-decoder segmentation network could be used. We visualise the features in the last convolutional blocks in each decoder.

each followed by ReLU and a normalisation layer. In order to learn the magnitude of feature change at each pixel, we apply a Sigmoid function at the end of the side branch. We use element-wise multiplication of the output of the side branch with the output of the main branch to perturb the features of the main branch. We then apply a skip-connection on the perturbed main branch output to yield the final output of the PASB.

### 2.2. Negative Attention Shifting Block in $f_{d2}$

The Negative Attention Shifting Block (Purple block in Fig.6) erodes foreground features using negative attention to the foreground. Following PASB, we design the NASB again as two parallel branches. The main branch is the same with the one in the PASB. The side branch is similar with the main branch but with a skip-connection on each convolutional layer. We also apply a Sigmoid function on the output of the side branch to learn the perturbation magnitude at each pixel. Then we multiply the learnt eroding attention mask from the side branch with the output of the main branch. We also apply a skip-connection on the perturbed main branch output to yield the final output of the NASB.

### 2.3. Loss Functions

We use a streaming training setting to avoid over-fitting on limited labelled data so the model doesn't repeatedly see the labelled data during each epoch. For labelled data we apply a standard Dice loss (Milletari et al., 2016) with the output of each decoder. For unlabelled data we apply a mean squared error loss between the outputs of the two decoders. This consistency regularisation is weighted by hyper-parameter $\alpha$ between 0.0005 to 0.004, which is also annealed during the training. Similar to SimSiam(Chen and He, 2021), we cut the gradients via detaching operation in Pytorch when applying the consistency regularisation on the two decoders.

## 3. Experiments

**CARVE 2014** The Classification of pulmonary arteries and veins (CARVE) dataset (Charbonnier et al., 2015) has 10 fully annotated non-contrast low-dose thoracic CT scans. Each case has between 399 and 498 images, acquired at various resolutions between (282 x 426) to (302 x 474). 10-fold cross-validation on the 10 labelled cases is performed. In each fold, we split cases as: 1 for labelled training data, 3 for unlabelled training data, 1 for validation and 5 for testing. We have more than 2000 slices for testing. We only used slices containing more than 100 foreground pixels. We prepared datasets with differing amounts of labelled slices: 5, 10, 30, 50, 100. It is worthy to mention the most 100 slices is equal to about 10% of the whole available labelled data. We cropped $176 \times 176$ patches from four corners of each slice. Full label training uses 4 training cases. Normalisation was performed at case wise.

    **BRATS 2018** BRATS 2018 (Menze et al., 2015) has 210 high-grade glioma and 76 low-grade glioma MRI volumes, each case containing 155 slices. We focus on binary segmentation of whole tumours in high grade cases. We randomly selected 1 case for labelled training, 2 cases for validation and 40 cases for testing. We have 6200 slices for testing. We centre cropped slices at $176 \times 176$. For labelled training data, we discarded empty slices and extracted the first 20 slices containing tumours with areas of more than 5 pixels. To see the impact of the amount of unlabelled training data, we used different numbers of slices at 3100 (20 cases), 4650 (30 cases) and 6200 (40 cases) respectively. Case-wise normalisation was performed and all modalities were concatenated.

    **Experimental Settings** We performed five sets of experiments/analysis: 1) comparisons with baselines including supervised learning and state-of-the-art semi-supervised learning approaches (Sohn et al., 2020; Tarvainen and Valpola, 2017; Chen et al., 2019; Ouali et al., 2020) using either data or feature augmentation; 2) investigation of the impact of the amount of labelled data and unlabelled data on MisMatch performance; 3) ablation study of the decoder architectures; 4) ablation study on the hyper-parameter, on the CARVE dataset using 5 labelled slices; 5) calibration analysis of MisMatch, cross-validation on the CARVE with 50 labelled slices.

### 3.1. Baselines

The backbone is a 2D U-net (Ronneberger et al., 2015) with 24 channels in the first encoder. To ensure a fair comparison we use the same U-net as the backbone across all baselines.

The first baseline utilises supervised training on the backbone, is trained with labelled data, augmented with flipping and Gaussian noise and is denoted as "Sup1". To investigate how unlabelled data improves performance, our second baseline "Sup2" utilises supervised training on MisMatch, with the same augmentation. Because MisMatch uses consistency regularisation, we focus on comparisons with five consistency regularisation SSL methods: 1) "mean-teacher" (MT) (Tarvainen and Valpola, 2017), with Gaussian noise, which has inspired most of the current state-of-the-art SSL methods; 2) the current state-of-the-art model called "FixMatch" (FM) (Sohn et al., 2020). To adapt FixMatch for a segmentation task, we use Gaussian noise as weak augmentation and "RandomAug" (Cubuk et al., 2020) without shearing for strong augmentation. We do not use shearing for augmentation because it impairs spatial correspondences of pixels of paired dense outputs; 3) a state-of-the-art model with multi-head decoder (Ouali et al., 2020) for segmentation (CCT), with random feature augmentation including Dropout(Srivastava et al., 2014), VAT(Miyato et al., 2017) and CutOut(DeVries and Taylor, 2017), et al. This baseline is also similar to models recently developed (French et al., 2020; Ke et al., 2020); 4) a further recent model in medical imaging (Chen et al., 2019) using image reconstruction as an extra regularisation (MTA), augmented with Gaussian noise; 5) a U-net with two standard decoders, where we respectively apply traditional erosion and dilation on the features directly in each decoder, augmented with Gaussian noise (Morph)". Our MisMatch model has been trained without any augmentation. See Appendix.A for details of training and implementation.

## 4. Results and Discussion

| Labelled | Supervised | | Semi-Supervised | | | | | |
|---|---|---|---|---|---|---|---|---|
| Slices | Sup1 | Sup2 | MTA | MT | FM | CCT | Morph | MM |
| 5 | 48.32±4.97 | 50.75±2.0 | 54.91±1.82 | 56.56±2.38 | 49.30±1.81 | 52.54±1.74 | 52.93±2.19 | **60.25±3.77** |
| 10 | 53.38±2.83 | 55.55±4.42 | 57.78±3.66 | 57.99±2.57 | 51.53±3.72 | 55.25±2.52 | 57.08±2.96 | **60.04±3.64** |
| 30 | 52.09±1.41 | 53.98±4.42 | 60.78±4.63 | 60.46±3.74 | 55.16±5.93 | 60.81±4.09 | 60.19±4.97 | **63.59±4.46** |
| 50 | 60.69±2.51 | 64.79±3.46 | 68.11±3.39 | 67.21±3.05 | 62.91±6.99 | 65.06±3.42 | 64.88±3.25 | **69.39±3.74** |
| 100 | 68.74±1.84 | 73.1±1.51 | 72.48±1.61 | 71.48±1.57 | 72.58±1.84 | 72.07±1.75 | 72.11±1.88 | **74.83±1.52** |
| Param. (M) | 1.8 | 2.7 | 2.1 | 1.88 | 1.88 | 1.88 | 2.54 | 2.7 |
| Infer.Time(s) | 4.1e-3 | 1.8e-1 | 7.2e-3 | 4.3e-3 | 4.5e-3 | 1.5e-1 | 8e-3 | 1.8e-1 |
| P values | 9.13e-5 | 1.55e-2 | 4.5e-3 | 4.3e-4 | 1.05e-2 | 1.8e-3 | 2.2e-3 | – |

Table 1: MisMatch (MM) vs Baselines on CARVE. Metric is Intersection over Union (IoU): mean (std) under 10-fold cross validation. P values from Mann-Whitney U-Test against MisMatch. Red: best model. Blue:2nd best model.

| Unlabelled | Supervised | | Semi-Supervised | | | | | |
|---|---|---|---|---|---|---|---|---|
| Slices | Sup1 | Sup2 | MTA | MT | FM | CCT | Morph | MM |
| 3100 | 53.74±10.19 | 55.76±11.03 | 50.53±8.76 | 55.29±10.21 | 57.92±12.35 | 56.61±11.7 | 53.88±9.99 | **58.94±11.41** |
| 4650 | 53.74±10.19 | 55.76±11.03 | 47.36±6.65 | 58.32±12.07 | 54.29±9.69 | 56.94±10.93 | 55.82±11.03 | **60.74±12.96** |
| 6200 | 53.74±10.19 | 55.76±11.03 | 50.11±8.00 | 56.92±12.20 | 56.78±11.39 | 57.37±11.74 | 54.5±9.75 | **58.81±12.18** |

Table 2: MisMatch (MM) vs Baselines on BRATS. Metric is Intersection over Union (IoU). Each model was trained 3 times. Red: best model. Blue:2nd best model.

**Segmentation Performance:** 1) MisMatch consistently and substantially outperforms supervised baselines, e.g. 24% improvement over Sup1 on 5 labelled slices, CARVE; 2) MisMatch consistently outperforms previous SSL methods (Sohn et al., 2020; Tarvainen and Valpola, 2017; Chen et al., 2019; Ouali et al., 2020) in Table 1, across different data sets, e.g. statistical difference when 6.25% labels (100 slices comparing to 1600 slices of full label) are used on CARVE (Table 1); 3) more labelled training data consistently produces a higher mean IoU and lower standard deviation (Table 2). Visual results can be found in Fig.8 in Appendix.E.

**Ablation Studies** We performed ablation studies on the architecture of the decoders of MisMatch (Fig3(a)) with cross-validation on 5 labelled slices of CARVE: 1) "MM-a", a two-headed U-net with standard convolutional blocks in decoders, this model can be seen as no feature perturbation, however, they are essentially slightly different because of random initialisation, we denote the decoder of U-net as $f_{d0}$; 2) "MM-b", a standard decoder of U-net and a negative attention shifting decoder $f_{d2}$, this one can be seen as between no perturbation and learnt erosion perturbation; 3) "MM-c", a standard decoder of U-net and a positive attention shifting decoder $f_{d1}$, this one can be seen as between no perturbation and learnt dilation perturbation; 4) "MM", $f_{d1}$ and $f_{d2}$ (Ours). As shown in Fig3(b), our MisMatch ("MM") outperforms other combinations in 8 out of 10 experiments and it performs on par with the others in the rest 2 experiments. We also tested $\alpha$ at 0, 0.0005, 0.001, 0.002, 0.004 with the same experimental setting. The optimal $\alpha$ appears at 0.002 in Fig.3(c).

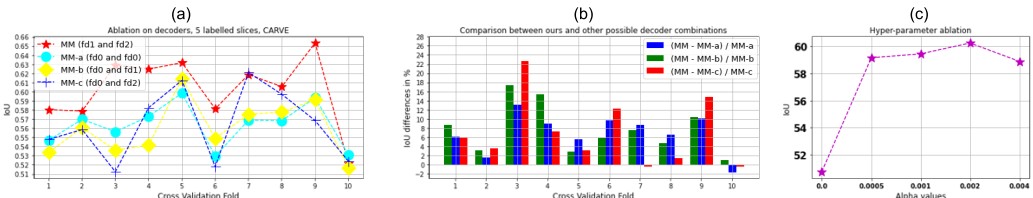

Figure 3: (a) Ablation studies on decoder architectures. (b) Performance differences between other possible decoder combinations against our used design. (c) Ablation study on the hyper-parameter alpha which weights the consistency loss. All experiments were performed on 5 labelled slices with CARVE with cross-validation

**MisMatch Is Better Calibrated** We conjugate that MisMatch utilises unlabelled images to improve model calibration (Guo et al., 2017), leading to better segmentation performance. Model calibration reflects the trustworthiness of the network predictions, which are crucial in clinical applications. Following (Guo et al., 2017), we set $B_m$ as the subset of all pixels whose prediction confidence is in interval $I_m$. We define accuracy as how many pixels are correctly classified in each confidence interval. The accuracy of $B_m$ is: $acc(B_m) = \frac{1}{|B_m|} \sum_{i \in B_m} 1(\hat{y}_i = y_i)$. Where $\hat{y}_i$ is the predicted label and $y_i$ is the ground truth label at pixel $i$ in $B_m$. The average confidence within $B_m$ is defined with the use of $\hat{p}_i$ which is the raw probability output of the network at each pixel: $conf(B_m) = \frac{1}{|B_m|} \sum_{i \in B_m} \hat{p}_i$. The plot for comparing the accuracy and confidence for each interval is called Reliability map (Fig.4), the gap between the accuracy and the confidence is called expected calibration error,

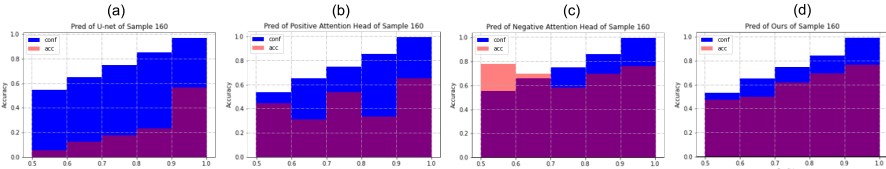

Figure 4: Reliability diagrams of one testing image. Blue: Confidence. Red: Accuracy. (a): Sup1 (Supervised learning with U-net). (b): Outputs of positive attention decoders. (c): Outputs of negative attention decoders. (d): Average outputs of the two decoders. The smaller the gap between the accuracy and the confidence, the better the network is calibrated.

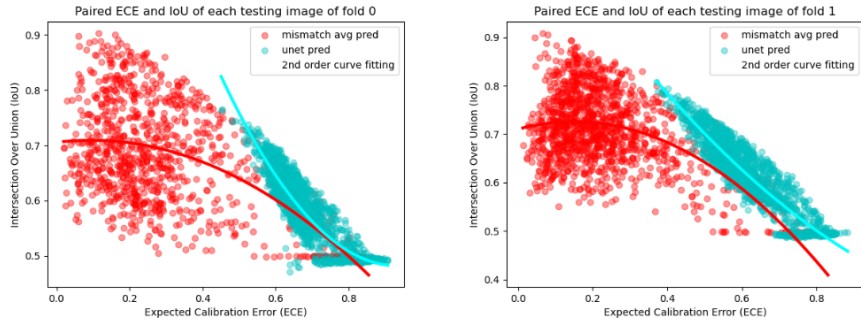

Figure 5: Expected calibration error (ECE) against segmentation performances (IoU) from cross-validation on CARVE with 50 labelled slices. $ECE = \sum_{m=1}^{M} \frac{|B_m|}{n} |acc(B_m) - conf(B_m)|$. The lower the ECE value, the better the model is calibrated.

the smaller the gap, the better the network is calibrated. As shown in the testing result in Fig.4 and Fig.5 from experiments trained on 50 labelled slices of CARVE, MisMatch produces better calibrated predictions.

## 5. Conclusion

We propose MisMatch, a consistency-driven SSL framework with attention-based feature augmentation for semi-supervised segmentation of medical images. MisMatch promises strong clinical utility by reducing the number of training labels required by more than 90%: when trained on just 10% of labels, MisMatch achieves a similar performance (IoU: 75%) to models that are trained with all available labels (IoU: 77%). Future work can further explore the calibration of models to understand why consistency regularisation works.

## Acknowledgments

We thank the anonymous reviewers for helping us to improve the paper quality with their constructive reviews. MCX is supported by GlaxoSmithKline (BIDS3000034123), EPSRC CDT in i4Health and UCL Engineering Dean's Prize. NPO is supported by a UKRI Future Leaders Fellowship (MR/S03546X/1). DCA is supported by UK EPSRC grants M020533, R006032, R014019, V034537, Wellcome Trust UNS113739. JJ is supported by Wellcome Trust Clinical Research Career Development Fellowship 209,553/Z/17/Z. NPO, DCA, and JJ are supported by the NIHR UCLH Biomedical Research Centre, UK.

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

## Appendix A. Implementation

We use Adam optimiser (Kingma and Ba, 2015). Hyper-parameters are: $\alpha = 0.002$, batch size 1 (GPU memory: 2G), learning rate 2e-5, 50 epochs. Each complete training on CARVE takes about 3.8 hours. The final output is the average of the outputs of the two decoders. In testing, we take an average of models saved over the last 10 epochs across experiments. Our code is implemented using Pytorch 1.0 (Paszke et al., 2019).

## Appendix B. Related Work

**SSL in classification** A recent review (Oliver et al., 2018) summarised different common SSL (Iscen et al., 2019) (Miyato et al., 2017) (Tarvainen and Valpola, 2017) methods including entropy minimisation, label propagation methods, generative methods and consistency based methods. Entropy minimisation encourages models to produce less confident predictions on unlabelled data (Grandvalet and Bengio, 2004) (Lee, 2013). However, entropy minimisation might overfit to the clusters of classes and fail to detect the decision boundaries of low-density regions (see Appendix E in (Oliver et al., 2018)). Label propagation methods (Iscen et al., 2019) (Lee, 2013) aim to build a similarity graph between labelled data points and unlabelled data points in order to propagate labels through dense unlabelled data regions. Nevertheless, label propagation methods need to build and analyse their Laplacian matrices which will limit their scalability. Generative models have also been

used to generate more data points in a joint optimisation of both classification of labelled data points and generative modelling (Kingma et al., 2014). However, the training of such a joint model can be complicated and unstable. On the other hand, consistency regularisation methods have achieved state-of-the-art performances across different benchmarks, additionally, consistency regularisation methods are simple and can easily be scaled up to large data sets. Of the consistency regularisation methods, Mean-Teacher (Tarvainen and Valpola, 2017) is the most representative example, containing two identical models which are fed with inputs augmented with different Gaussian noises. The first model learns to match the target output of the second model, while the second model uses an exponentially moving average of parameters of the first model. The state-of-the-art SSL methods (Berthelot et al., 2020) (Sohn et al., 2020) combines two categories: entropy minimisation and consistency regularisation.

**SSL in segmentation** In semi-supervised image segmentation, consistency regularisation is commonly used (Xu et al., 2020a) (Li et al., 2020) (Cui et al., 2019) (Hang et al., 2020) (Fang and Li, 2020) (French et al., 2020) where different data augmentation techniques are applied at the input level. Another related work (Li et al., 2018) forces the model to learn rotation invariant predictions. Apart from augmentation at the input level, recently, feature level augmentation has gained popularity for consistency based SSL segmentation (Ouali et al., 2020; Ke et al., 2020). Apart from consistency regularisation methods in medical imaging, there also have been other attempts, including the use of generative models for creating pseudo data points for training (Chaitanya et al., 2019) (Chen et al., 2020) and different auxiliary tasks as regularisation (Kervadec et al., 2019) (Chen et al., 2019). Since our method is a new consistency regularisation method, we focus on comparing with state-of-the-art consistency regularisation methods.

## Appendix C. Detailed Architectures of Blocks of MisMatch

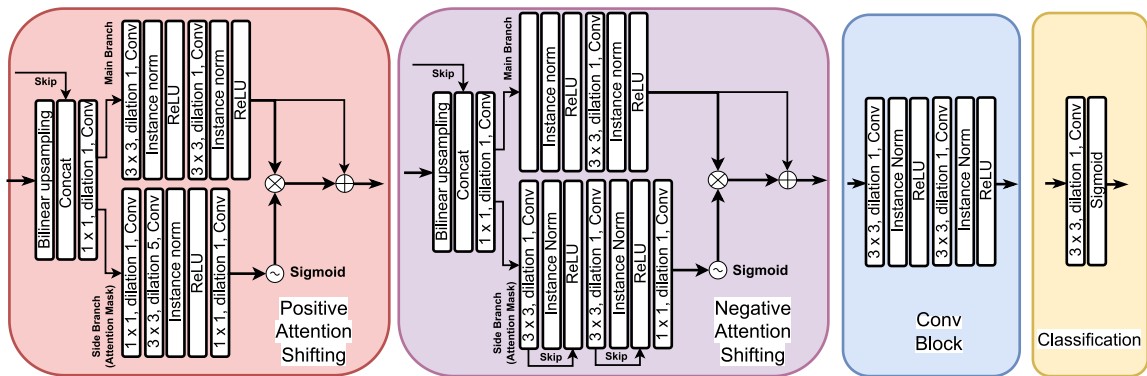

Figure 6: $f_e$ has 3 convolutional blocks. $f_{d1}$ has 3 PASBs followed by a classification layer. $f_{d2}$ has 3 NASBs followed by a classification layer.

## Appendix D. More Reliability Maps

See Fig.7.

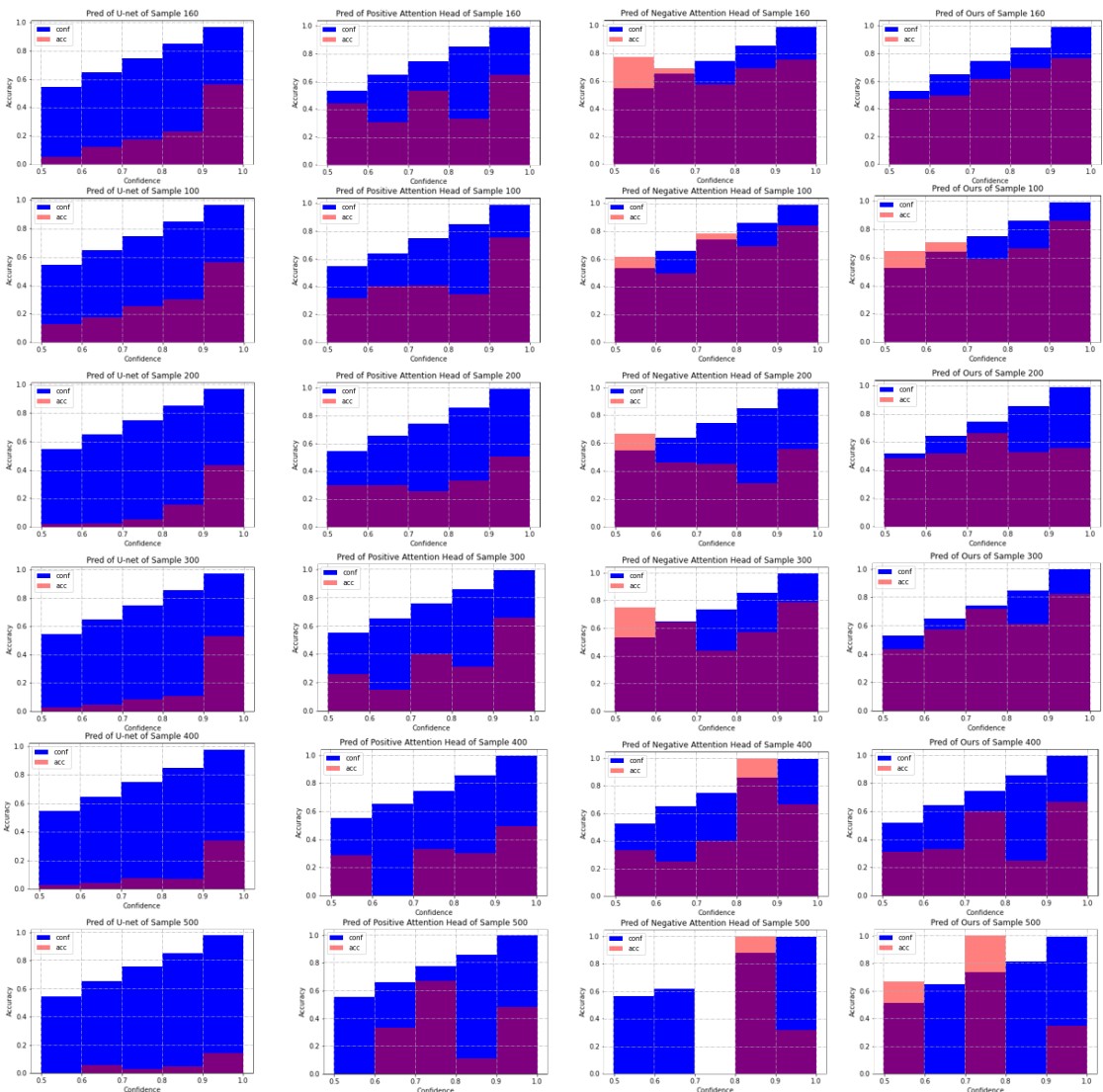

Figure 7: More reliability maps. Blue: Confidence. Red: Accuracy. Each row is on one testing image. X-axis: bins of prediction confidences. Y-axis: accuracy. Column 1: Sup1 (supervised learning with U-net). Column 2: outputs of positive attention decoders. Column 3: outputs of negative attention decoders. Column 4: average outputs of the two decoders. The smaller the gap between the accuracy and the confidence, the better the network is calibrated.

## Appendix E. Visual Results

See Fig.8.

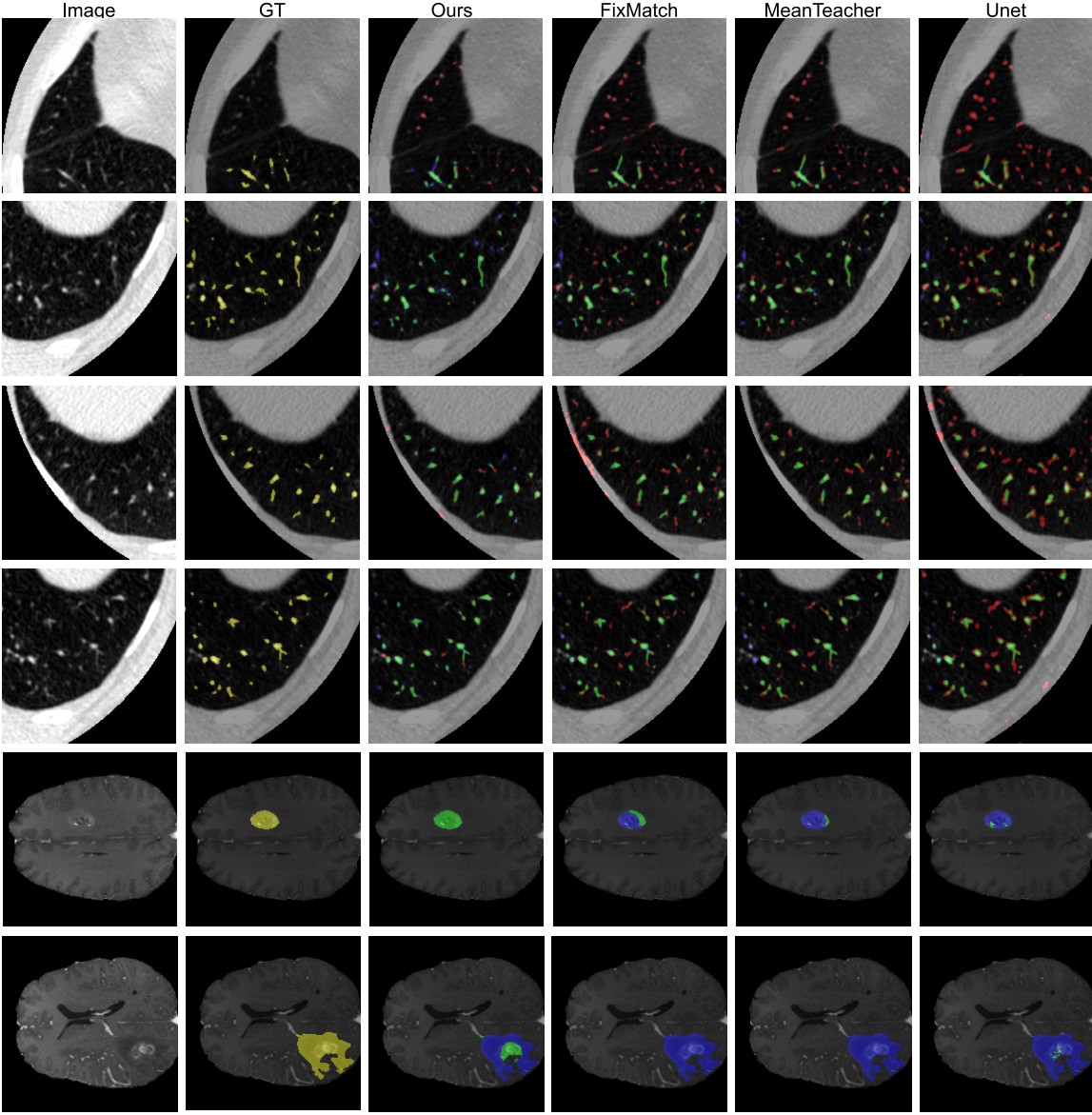

Figure 8: Visual results. Row 1-4: CARVE. Row 5-6: BRATS. Yellow: ground truth. Red: False Positive. Green: True Positives. Blue: False Negatives.

