# OpenReview forum: "Learning Morphological Feature Perturbations for Calibrated Semi-Supervised Segmentation"
_MIDL.io/2022/Conference — MIDL 2022_

### Official Review · Reviewer_y1rX · 2022-01-21

**Confidence:** 4
**Preliminary Rating:** 3
**Recommendation:** Poster

**Summary:**

The paper proposes a semi-supervised segmentation technique based on morphological feature perturbations. The method is assessed on two datasets (CT pulmonary vessel segmentation and MRI brain tumour segmentation). They compare their proposal against different supervised and semi-supervised techniques in the literature. The authors report superior results against these other techniques.

**Strengths:**

The papers seek to leverage morphological methods to improve segmentation, which is a relatively unexplored field. The authors report significant improvements in the two benchmarks they tested their method.

**Weaknesses:**

In my opinion, the greatest weakness of the paper is the clarity of the paper. The paper is long (16 pages), many figures are small and have poor quality, it is especially hard to discern some of the plots if you are colour blind.

There seems to be confusion about what the dilation and erosion morphological operators actually do. They are based on structuring elements, and for the gray-scale image case, they work with the supremum and infimum operations. The authors use Atrous convolutions and skip-connections, which is something different but seems to work well in the authors' experiments. That confusion made it harder for me to understand the methodology of the paper.




**Deanonymize Review:**

no

**Detailed Comments:**

- The authors report many results from experiments and ablation studies. It is easy to get confused. I would suggest keeping only the necessary results that support the authors' claims in the main manuscript. The rest could go to an appendix.

- Quality of some figures, especially some plots that are too small and use many different colours, could be improved.

- Too many acronyms, which often are not meaningful (e.g., sup1, sup2,...)

- What do the red and blue colours represent in the results tables?

- The proposed model has almost 50% more parameters than the Mean-Teacher model, for example. Isn't this an unfair comparison?

- why does the standard deviation of the inference time is not reported?

**Final Rating After The Rebuttal:**

4: Weak Accept

**Justification Of The Final Rating:**

I am mostly satisfied with the authors' responses, the new organization of the paper and the fact that the code is now publicly available. I updated my final rating to "weak accept". I think combining morphology with deep learning is a relatively unexplored field that is worth exploring.

**Paper Type:**

methodological development

**Questions To Address In The Rebuttal:**

- Please clarify the difference between morphological erosion and dilation and what the authors are using (skip-connections and Atrous convolutions). They are not the same thing.

- Please explain the red and blue colours in the results tables.

- Please explain what steps will be taken to make the manuscript more clear.


**Special Issue:**

no

---

### Official Review · Reviewer_FdRT · 2022-01-24

**Confidence:** 5
**Preliminary Rating:** 4
**Recommendation:** Poster

**Summary:**

The paper presents a new semi-supervised segmentation method introducing an encoder with 2 decoders architecture; the 2 decoders are designed to be specialised in foreground and background detection taking advantage of different dilated convolutions strategies. The method is evaluated on 2 different datasets, and they compare the proposed method with other methods from the literature.

**Strengths:**

I found the core idea (2 decoders specialised to perform 2 different tasks) very interesting, and I overall like the paper. The made a great work in comparing their methods with others and the experimental section is well designed. The paper is well written.

**Weaknesses:**

I detailed the paper weaknesses point by point in the section below. I feel there are 3 major points that should be addressed. If authors solve them, I see no reasons to reject the paper.
1. I recognise that could be tricky to obtain, but it would be great to have a better visualisation/explanation on the different branches, to better capture the different processing done by the main and side branches. In other words, to visualise how the two consecutive dilated convolutional layers create different results. In addition, it would be great to determine the contribution of both side branches. What happens if you don’t use them (visually)? What they produce? Can you say/show more on the perturbation provided by the side branch and multiplied with the main branch result? Keeping in mind, that you apply a skip-connection on the perturbed main branch. I feel it could be very interesting, if not critical to understand the method, to display this. I see that Figure 6 try to provide more insights on what it is happening for the two branches, but I found the figure very confusing. What histograms are supposed to prove? Can you overall add more clarity to the analysis? I feel the material is there, but the presentation could be improved.
2. Not sure why you used BRATS 2018, with a total of 542 MR scans, when BRATS 2021 has 2,040 patients improving a lot the quality and quantity of the data. See https://arxiv.org/pdf/2107.02314.pdf. Adding data from multiple institutions, they increased the difficulty since the distribution shift is one of the main problems when working with MRI data. This could actually more beneficial in your case, when you can have more unlabelled data to test your model. It is also very important to specify if training/val/testing sets are from the same MR site or not.
3. As a personal preference, I would suggest inserting figures in vector space rather than image space, so that it is possible to zoom and have better quality. Some Figures, like Fig.3, are very hard to appreciate and look worse than they should. Overall, I feel more explanations and better figures could give a huge boost to the paper.

**Deanonymize Review:**

no

**Detailed Comments:**

See section below

**Final Rating After The Rebuttal:**

4: Weak Accept

**Justification Of The Final Rating:**

I'm glad that authors spent the little amount of the rebuttal time to answer my questions and address my points.
I would go for the acceptance, although I think that Figure 6 and 7 should be in the paper, since they are crucial and they represent a cornerstone to explain the concept of the paper. But not in the form they are now, since they are pretty much the same as before, only bigger. Also, Figure 4 and 5 are still pretty ugly, maybe a restyling can help.

But despite that, I suggest acceptance.

**Paper Type:**

methodological development

**Questions To Address In The Rebuttal:**

1.	In the data description section (3. Experiments), what do you mean with MRI cases? Volumes? Patients type?
2.	In table 1 and 2, please specify in the caption what red and blue colours means.
3.	Can you please specify which type of regularisation did you used to control the supervised training? Since you used only 5 slices, the chances to overfit the training are concrete. So, it would be interesting to know (in Supp. maybe) which strategies you used to prevent it.
4.	Figure 2, please specify x axis labels (slices, right?). The entire caption can be improved a little bit (I think all of them, can be improved).
5.	In the sentence “MisMatch consistently outperforms previous SSL methods (Sohn et al., 2020; Tarvainen and Valpola, 2017; Chen et al., 2019; Ouali et al., 2020) in Table 4, across different data sets, e.g. statistical difference when 6.25% labels (100 slices comparing to 1600 slices of full label) are used on CARVE (Table 4)”
You state Table 4, but the link is to Section “4. Results and Discussion”. Please correct. Same problem in the first and second lines of page 6.
6.	Pag. 5, the statement “As shown in Fig 4, the main performance boost of MisMatch comes from the reduction of false positive detection and the increase of true positive detection;” it could benefit from ad-hoc analysis, adding a quantitative analysis in the supplementary (like a table with TP, FN, etc. is enough). At least more data to support the statement.
7.	The sentence “5) more unlabelled training data can help with generalisation, until it dominates training and impedes performance” is not very clear. Does it mean that performance decay if too much unlabelled training data are used? Where do you see that?
8.	Very important the Ablation Studies and well designed. Not sure why you decided to go for the 5 slices settings, since results are less consistent and robust than 50 or 100 slices.
9.	Figure 3: In addition to (or in replacement of) different cross validation folds, is it possible to summarise across different folds? With mean/std. Please change yellow line, it’s not that easy to watch. Also, please name 3-a, 3-b, 3-c. It’s easier to link them in the manuscript. The caption is not really informative, please add more (especially on the figure in the middle, that it is not commented in the text).
10.	Page 6 “We also tested α”. What alpha is? If it is an hyperparameter optimisation, then why it is in the ablation study section? Please move it in an appropriate section.
11.	Figure 5, Caption. What “Column 1: U-net” is supposed to be? The supervised settings?
12.	To be honest, I didn’t fully understand the analysis “MisMatch Is Better Calibrated” and “Robustness of MisMatch Against Expected Calibration Errors”. It could be me, or simply it could be useful to add a bit more explanations in the text.
13.	A question arises reading the experiment sections. What do you think results will be, in those situations where there is not a clear advantage of foreground-background separation? In other words, if the task is not segmenting small portion of the image (i.e., the background class covers the most of the image), but with multiple classes covering the entire space, do you expect same results?
14.	I like a lot the Conclusions. You write “MisMatch promises strong clinical utility by reducing the number of training labels required by more than 90%: when trained on just 10% of labels, MisMatch achieves a similar performance (IoU: 75%) to models that are trained with all available labels”. That is a direct message that I cannot read in that along the paper. Using percentages i.e., saying only 10% of labels are needed, it is something I don't see in the paper but that could be a better way than saying “we used 5-10 slices”. Is it possible to show this in the manuscript?

**Special Issue:**

no

---

### Official Review · Reviewer_fftq · 2022-01-24

**Confidence:** 4
**Preliminary Rating:** 3
**Recommendation:** Poster

**Summary:**

The paper proposes a semi-supervised segmentation techniques which uses so-called consistency regularisation via different morphological feature dilations to increase performance with unlabelled data. The method is evaluated on two medical imaging datasets BRATS 2018 and CARVE 2014. The authors claim it outperforms state-of-the-art semi-supervised methods on CT pulmonary vessel segmentation and MRI brain tumor segmentation.

**Strengths:**

- The work addresses the highly relevant and important topic of learning segmentation models with limited data.

- The method shows consistently good results in comparison to supervised and other semi-supervised approaches. All recent and relevant.

- The paper is generally well written, only has a few minor problems with structure, and as far as I know adequately references prior work.


**Weaknesses:**

- 2D processing of 3D data. I have a hard time seeing the relevance of relatively complicated architecture and training experiments with limited subsets of 2D slices pulled from 3D data. How can we trust that the methods can be extended to and results generalize with 3D processing and/or in comparison with state of the art 3D methods. Why not work with actual 2D data then?

- Complicated and hard to follow method design that is barely motivated. Please clarify - Why focus on dilation and erosion of foreground features? Are these operations particular useful and why? How and why did you arrive at the architecture blocks described in section 2.1 and 2.2?

- Datasets are limited and not used in matter that makes comparison with prior works easy. E.g. we are told state-of-the-art performance is achieved, but in reality the authors have trained all the comparison methods themselves and with dataset splits and subsets that are as far as I am aware unique to this work. So the results are unlikely to be directly comparable with prior and coming works.

**Deanonymize Review:**

no

**Detailed Comments:**

- Appendix B - I do not really follow the link between low density regions in the data distribution at different levels, e.g. images and features and the need for pertubations at these levels.

- What is compared with the p-values in table 1?

- The results appear to incorrectly refer to Table 4 and not Table 1.

- "To investigate how unlabelled data improves performance, our second baseline “Sup2” utilises supervised training on MisMatch, with the same augmentation.", I am not sure what you mean by supervised training on MisMatch.

- In Figure 2, please use same method abbreviations as elsewhere. Describe what is on the x-axis. Remove dots from the full label line, as I understand it is actually not trained with these data splits, so it is misleading to add points.

- In Figure 5, please use same method abbreviations as elsewhere. I don't understand the reliability diagrams from this figure. Figure 8 in the appendix comes a bit closer. Consider being more detailed in the caption.

- I think there is too much going on in figure 5 and figure 6. The text is not readable and captions are unable to clearly tell the story because there is so much content.

- Please swap position of figure 5 and 6 so that they are closer to where they are referenced.

- I find the Covid-19 point in the conclusion a bit detached and irrelevant. Being able to learn to segment images with less labelled data is obviously relevant in a multitude of applications.


**Final Rating After The Rebuttal:**

4: Weak Accept

**Justification Of The Final Rating:**

I would like to thank the authors for their substantial revision of their manuscript and I am happy by the way my comments were addressed. I have chosen to change my recommendation to weak accept. While the results are not positive for the proposed augmentations as a whole, the reporting is now more accurate.

**Paper Type:**

methodological development

**Questions To Address In The Rebuttal:**

- The introduction seems to indicate that there is a difference between semi-supervised classification and segmentation in that the latter requires feature-level pertubations. Is segmentation really different from classification in this respect? It seems to me that both classification and segmentation can benefit from both input and feature-level pertubations. I have a hard time following the explanation in the Appendix B. See comments elsewhere.

- So many new methods are introduced these days and very often they show good results that are not replicated by future works. As also mentioned above, I think there is substantial task in motivating the choices made. This may require a more streamlined focus on the main points. Will this be done and how?


**Special Issue:**

no

---

### Meta-Review · Area_Chair_3Z9U · 2022-02-18

**Recommendation:** Accept (Poster)
**Confidence:** 4

**Metareview:**

The paper proposes a semi-supervised segmentation technique which uses consistency regularization via different morphological feature perturbations to increase performance with unlabelled data.  It consists of an encoder and two-head decoders. One decoder learns positive attention to the foreground regions generating dilated features using atrous convolutions. The other decoder learns negative attention to the foreground on the same image thereby generating eroded features using skip connections. A consistency regularization is then applied on these paired predictions of an unlabeled image to improve performance. The method is evaluated on two different datasets, and the authors compared the proposed method with other methods from the literature, and reported superior results.

During the rebuttal phase the authors made significant efforts to revise their manuscript especially to improve clarity of their presentation and after rebuttal, all reviewers agreed for weak acceptance. I also agree with this decision and support publication of this paper.

---

### Decision · Program_Chairs · 2022-02-28

Accept